# Metabolism of the Genus *Guyparkeria* Revealed by Pangenome Analysis

**DOI:** 10.3390/microorganisms10040724

**Published:** 2022-03-28

**Authors:** Maggie C. Y. Lau Vetter, Baowei Huang, Linda Fenske, Jochen Blom

**Affiliations:** 1Laboratory of Extraterrestrial Ocean Systems, Institute of Deep-sea Science and Engineering, Chinese Academy of Sciences, Sanya 572000, China; huangbw@idsse.ac.cn; 2University of Chinese Academy of Sciences, Beijing 100049, China; 3Bioinformatics & Systems Biology, Justus Liebig University Gießen, Heinrich-Buff-Ring 58, 35390 Gießen, Germany; linda.fenske@computational.bio.uni-giessen.de (L.F.); jochen.blom@computational.bio.uni-giessen.de (J.B.)

**Keywords:** *Guyparkeria*, pangenome analysis, metabolic capability, adaptation

## Abstract

Halophilic sulfur-oxidizing bacteria belonging to the genus *Guyparkeria* occur at both marine and terrestrial habitats. Common physiological characteristics displayed by *Guyparkeria* isolates have not yet been linked to the metabolic potential encoded in their genetic inventory. To provide a genetic basis for understanding the metabolism of *Guyparkeria*, nine genomes were compared to reveal the metabolic capabilities and adaptations. A detailed account is given on *Guyparkeria*’s ability to assimilate carbon by fixation, to oxidize reduced sulfur, to oxidize thiocyanate, and to cope with salinity stress.

## 1. Introduction

The halophilic sulfur-oxidizing species *Guyparkeria* [1] was previously classified as a member of *Thiobacillus* [2] and then of *Halothiobacillus* [3]. Currently, two species are declared, namely the type strain *G. halophila* DSM 6132 (which is also the type species of *Guyparkeria* [4]) and the type strain *G. hydrothermalis* R3 [2]. Since then, more *Guyparkeria* isolates and metagenome-assembled genomes (MAGs) have been identified. They originated from marine and terrestrial environments: hydrothermal vent chimney, deep-sea sediment, coastal sediment, hot spring sediment, and lacustrine sediment (see Table 1 for details). *Guyparkeria* is also being explored to be used as a more environmentally friendly strain in metal extraction from ores [5]. As of now, common physiological characteristics displayed by *Guyparkeria* isolates have not been linked to the metabolic potential encoded in their genetic inventory. The genome of strain SCN-R1 has been studied [6,7], but the focus mainly centered around its thiocyanate oxidation, an ability that is so far exclusive to this strain.

Seven genomes assigned to *Guyparkeria* based on the Genome Taxonomy Database (GTDB) [8] taxonomic classification have become available in the past few years, making it possible to perform a genome-wide study. To provide a genetic basis for understanding the metabolism of the genus *Guyparkeria*, genomes were sequenced from *G. hydrothermalis* R3, isolated from hydrothermal vent chimney [2], and a coastal strain B1-1 [9] and compared with the seven publicized genomes. The metabolic capabilities and adaptations are discussed on the basis of the pan genome data.

## 2. Materials and Methods

### 2.1. Microorganisms and Culturing Media

*G. hydrothermalis* R3 (hereafter, R3) was isolated from an active hydrothermal vent chimney at 2000 m depth at the North Fiji Basin, Tonga, Pacific Ocean [2]. Live culture of R3 was purchased from DSMZ—German Collection of Microorganisms and Cell Cultures GmbH (catalog number: 7121). *G. hydrothermalis* B1-1 (hereafter, B1-1) was isolated more recently from sediments of marine culture cage area at Bachimen, Zhangzhou, Fujian [9]. B1-1 was determined to the species level based on its high 16S rRNA gene similarity (>99%) and phylogenetic relatedness to R3 [9]. Live culture of B1-1 was purchased from China Center for Type Culture Collection (catalog number: CCTCC AB 2016151).

Live culture of R3 cells was maintained at 35 °C on 1% agar plates prepared from DSMZ medium 574, with minor modifications, containing (per L) 25 g NaCl, 2.5 g Na_2_S_2_O_3_·5H_2_O, 1 g (NH_4_)_2_SO_4_, 1 g anhydrous MgSO_4_, 0.4 g K_2_HPO_4_, 0.2 g NaHCO_3_, 0.23 g anhydrous CaCl_2_, 200 mL Tris-HCl buffer (0.1 M, pH 7.5), 800 mL sterile distilled water and 1 mL of trace element solution (per L of sterile distilled water: 50 g Na_2_-EDTA, 2.2 g ZnSO_4_·7H_2_O, 4.15 g anhydrous CaCl_2_, 3.96 g anhydrous MnCl_2_, 5 g FeSO_4_·7H_2_O, 1.1 g (NH_4_)_6_Mo7O_24_·4H_2_O, 0.81 g anhydrous CoCl_2_, adjusted to pH 6.0 using 5 M NaOH), adjusted to pH 7.5. The medium was filtered twice through a 0.22-μm-pore-size membrane filter for sterilization.

Live culture of B1-1 cells was maintained at 28 °C on 1% agar plates prepared from a medium, following Chen et al. [9] with minor modifications, containing (per L) 0.05 g anhydrous MgSO_4_, 5 g Na_2_S_2_O_3_·5H_2_O, 2 g K_2_HPO_4_, 0.1 g (NH_4_)_2_SO_4_, 0.08 g anhydrous CaCl_2_, 0.02 g FeSO_4_·7H_2_O in 1 L of aged seawater (i.e., sea water collected from the nearby coast and was kept in dark for at least 1 week before use), adjusted to pH 7.6. The medium was sterilized by autoclaving.

Purity of R3 and B1-1 cultures was ensured by making multiple transfers from a single colony using streak plate method. The near-full length 16S rRNA gene was amplified from single colonies by polymerase chain reaction (PCR): a 50 μL reaction mixture containing 25 μL 2× Glfex Mix (Takara Biomedical Technology (Beijing) Co., Ltd., Beijing, China), 2.5 μL forward and reserve primers (1 μM; BGI Genomics, Guangzhou, China), 0.1–10 μL cell suspension, and distilled water (autoclaved and filter-sterilized), and then was subjected to denaturation at 95 °C for 5 min, 30 cycles of denaturation at 95 °C for 30 s, annealing at 55 °C for 30 s and extension at 72 °C for 90 s, and a final extension at 72 °C for 5 min on the T30 thermal cycler (Hangzhou LongGene Scientific Instruments Co., Ltd., Hangzhou, China). Primers 27F (5′-AGAGTTTGATCCTGGCTCAG-3′) and 1492R (5′-TACGGYTACCTTGTTACGACTT-3′) were used. PCR products were purified by ethanol precipitation, checked by gel electrophoresis in 1X Tris-Acetate-EDTA buffer (50× solution of 2 M Tris base, 0.1 M Na_2_EDTA·2H_2_O and 1 M glacial acetic acid) at 120 V for 30 min, and quantified using a nanodrop instrument (model Nano-500, Hangzhou Allsheng Instruments CO., Ltd., Hangzhou, China) before sending to BGI Genomics (Guangzhou, China) for Sanger sequencing.

### 2.2. Genomic DNA Extraction and Sequencing of Strains B1-1 and R3

Batch culture (3–6 mL) was centrifuged at 10,000 rpm for 1 min and the supernatant was discarded. The cell pellet was then suspended in 180 μL of lysozyme solution (20 mg/mL lysozyme in buffer containing 20 mM Tris, pH 8.0; 2 mM Na_2_-EDTA; 1.2% Triton), and then incubated at 37 °C for 30 min. Cell lysis was achieved by added 400 μL of lysis buffer (containing 40 mM Tris; 20 mM sodium citrate; 1% SDS; 1 mM EDTA; 20 mg/mL proteinase K). After mixing with 300 μL of 5 M NaCl by inverting up-and-down., chloroform (800 μL) was added and mixed by inversion, then let the solution sit still for 2 min before centrifugation at 12,000 rpm for 10 min. About 500–600 μL of crude DNA solution was taken from the upper aqueous layer, to which double volume of absolute ethanol was added, before being left to sit for 2 min at room temperature to precipitate the DNA. After centrifugation at 12,000 rpm for 10 min, supernatant was discarded, and the tube was placed upside down to air dry. Finally, DNA pellet was suspended in 300 μL of sterile double-distilled water and the DNA solution was kept at −20 °C. DNA samples were sent to Novogene Co., Ltd. (Tianjin, China) at which metagenomic library was prepared using NEB Next Ultra DNA Library Prep Kit and sequenced by Illumina Novaseq 6000.

### 2.3. Quality Control and Assembly of Sequencing Data

Raw data (6,134,656 PE150 reads for strain B1-1, and 6,121,959 PE150 reads for strain R3) were filtered for quality using fastp v0.20.1 [10]: reads with 90% of bases with Qscore > 30 were kept, bases with Qscore < 25 (a sliding window of 4 nt) were removed from the 3′-end, and reads with no N and are >50-nt-long were kept (command: -q 30 -u 10 --cut_right --cut_right_window_size 4 --cut_right_mean_quality 25 -n 0 -l 50). With 9.7% and 13.2% of reads removed from B1-1 and R3 data, respectively, the remaining good paired reads were assembled for B1-1 and R3 individually using Unicycler v0.4.8 [11] and options --no_correct and --no_rotate. The assembled genomes were assessed by CheckM v1.0.12 [12].

### 2.4. Gene Annotation of Draft Genome B1-1 and R3

Draft genomes of B1-1 and R3 were first annotated using PROKKA v1.14.6 (default settings; [13]) and RAST (https://rast.nmpdr.org/, v2.0; accessed on 17 January 2022), and subsequently using Bakta v1.2.4 [14], and PGAP in NCBI submission portal [15]. For each genome, the four annotation files were converted from gbk format to gff3 format using an online tool (https://www.hiv.lanl.gov/content/sequence/FORMAT_CONVERSION/form.html; accessed on 17–19 January 2022) and manually compiled into a single file based on the gene start and end positions. Amino acid sequences of proteins predicted and annotated by PGAP were submitted to KEGG Automatic Annotation Server v2.1 (KAAS; [16]) to obtain KEGG Orthology and pathway assignments. BLAST search was performed against 40 selected gammaproteobacterial genomes (Appendix A), and bi-directional best hit method with default bit score thresholds (≥60) was used to assign orthologs. To identify transporter families, the PGAP annotated amino acid sequences were also searched against the TransportDB database (http://www.membranetransport.org/; accessed on 25 January 2022).

### 2.5. Pangenome and Core Genome Analyses

Seven genome sequences of *Guyparkeria* (Appendix A), as classified by GTDB in both the previous Release 95 and the current Release 202, were retrieved from the NCBI FTP site. It is noted that the taxonomy of these genome sequences in NCBI differs from that in GTDB, and only two (*Guyparkeria halophila* sp2 and *Guyparkeria* sp. SCN-R1) are classified as *Guyparkeria* by NCBI (Appendix A). Scaffolds shorter than 1500 nucleotides (nt) were discarded.

EDGAR v3.0 platform [17] was used to perform comparative genomic analysis and visualize the results. The tools used in this study include calculating the Average Amino Acid Identity (AAI) and Average Nucleotide Identity (ANI) values, assigning orthologous genes and using those to calculate the pangenome. EDGAR v3.0 was also used to derive exponential decay functions that predict the development of the pan genome or core genome with increasing genome number. The formulas for Heaps’ Law (the least squares fit of the power law to medians) for the pan genome (1) and the least squares fit of the exponential regressing decay to medians (2) were used, respectively [18]:*n* = *kN**^γ^*,(1)
where *n* is the number of genes, *N* is the number of genomes, and *k* and *γ* are fitted constants. α is defined as 1 − *γ*.
*n* = *ke^−N^*^/^*^τ^* + *tg**θ*,(2)
where *n* is the number of genes, *N* is the number of genomes, *e* is Euler number, and *k*, *τ* and *tg**θ* are constants.

Of the nine genomes included in this study, *G. halophila* sp2 genome is declared as a complete genome, therefore, it was used as the reference genome when analyzing core genes for *Guyparkeria*.

As genes encoding hydrogenase, thiosulfate:quinone oxidoreductase and thiosulfate dehydrogenase were not identified in the automated genome annotations, additional search using HMMER v3.3.2 (hmmer.org; installed on 26 February 2021) against specific HMM (hidden Markov model) profiles was performed using the command *hmmsearch*. The following HMM profiles were downloaded on 23 February 2022 from ftp://ftp.genome.jp/pub/db/kofam/: K23548 for [FeFe]-hydrogenase small subunit [EC 1.12.99.6]; K17992, K17993 and K17994 for [NiFe]-hydrogenase [EC EC:1.12.1.3]; K05908 for thiosulfate:quinone oxidoreductase [EC:1.8.5.2]; and K19713 for thiosulfate dehydrogenase [EC:1.8.2.2]. These HMM profiles were selected for their high quality, indicated by a F-measure value of greater than 90. Hits with a bit-score surpassing the recommended threshold values (293.67 for K23548; 114.10 for K17992, 375.77 for K17993 and 228.43 for K17994; 256.70 for K05908; and 200.50 for K19713) were considered valid.

### 2.6. Phylogenetic Analyses

16S rRNA genes were extracted from the nine genomes. They were missing in the genome of *Halothiobacillaceae* bacterium SpSt-1134. Sequences were aligned using MUSCLE v3.8.31 [19] in JalView v1.0. The multiple sequence alignment was analyzed by jModelTest v2.1.10 [20], and the recommended evolutionary model HKY+G was used to obtain a maximum likelihood tree followed by bootstrap analysis (––bs-trees autoMRE) using RAxML-NG v1.0.2 (httpes://github.com/amkozlov/raxml-ng; installed on 9 July 2021). 16S rRNA genes of *Halothiobacillus kellyi*, *Halothiobacillus neapolitanus*, *Thiovirga sulfuroxydans* and *Allochromatium vinosum* DSM 180 were used as outgroup.

Similarly, *sox*B genes were extracted from the nine genomes. They were missing in the genomes of *Halothiobacillaceae* bacterium SpSt-1134 and *Guyparkeria* sp. SCN-R1. Sequences were aligned using MUSCLE v3.8.31 in JalView v1.0. The multiple sequence alignment was analyzed by Prottest v3 [21], and the recommended evolutionary model LG+G was used to obtain a maximum likelihood tree followed by bootstrap analysis (––bs-trees autoMRE) using RAxML-NG v1.0.2. Similar to the 16S rRNA gene tree, soxB genes of *Halothiobacillus kellyi*, *Halothiobacillus neapolitanus*, *Thiovirga sulfuroxydans* and *Allochromatium vinosum* DSM 180 were used as outgroup.

With the exclusion of *Halothiobacillaceae* bacterium SpSt-1134 genome, 1458 core genes were extracted from the *Guyparkeria* genomes and aligned individually on the EDGAR v3.0 platform. Multiple sequence alignment file of concatenated core genes (in total 501,819 residue positions), provided by the EDGAR team, was analyzed by Prottest v3 and the recommended evolutionary model JTT+G was used to obtain a maximum likelihood tree followed by bootstrap analysis (––bs-trees autoMRE) using RAxML-NG v1.0.3 (httpes://github.com/amkozlov/raxml-ng; installed on 17 November 2021).

## 3. Results and Discussion

### 3.1. Basic Summary of Guyparkeria Genomes

The nine genomes are comprised of 1–235 scaffolds, with lengths of 1,188,407–2,594,469 nt (Table 1). CheckM estimated that they were 100% complete except three, namely *Halothiobacillus* sp. SB14A (99.43%), *Halothiobacillus* sp. S21.Bin061 (92.53%), and *Halothiobacillaceae* bacterium SpSt-1134 (68.91%). Only the genome of *Halothiobacillaceae* bacterium SpSt-1134 was estimated to have a very low level of contamination (1.15%). Thus, eight (out of nine) genomes were of very high quality.

The average GC content was 64.2 ± 4.5%. Percent of GC in the type strain *G. hydrothermalis* R3 genome was 66.5%, which is lower than the empirically determined values (67.1–68.3%) [2].

The numbers of coding sequences (CDS) predicted by CheckM and PROKKA for each genome were different. These differences were generally small (0–3.6%), given the total number of predicted CDS, except for *Halothiobacillaceae* bacterium SpSt-1134, for which the difference accounted for ~18%. Side-by-side annotation results for strains B1-1 and R3 are provided in Appendix A. As the annotations by NCBI PGAP and BAKTA yielded a lower percentage of hypothetical proteins, they were desirably used for the later comparative genomic analysis.

*G. hydrothermalis* B1-1 and R3 and *G. halophila* sp2 have two copies of 16S, 5S, 23S ribosomal RNA (rRNA) genes. *G. hydrothermalis* B1-1 and R3 genomes have identical 16S rRNA gene copies, whereas the two 16S rRNA genes in *G. halophila* sp2 genome are not identical, with one being more related to B1-1 and the other to R3 (Figure 1). The slight difference (3 out of 1539 nucleotide positions) between the two 16S rRNA gene copies in *G. halophila* sp2 genome could be a result of: (1) mixed populations or variants; (2) sequencing error; (3) assembly error; and (4) point mutation. With 16S rRNA gene pairwise identities between 96.46% and 100%, and AAI values >82.36% (Figure 2), the genomes compared, except *Halothiobacillaceae* bacterium SpSt-1134, can confidently be regarded as being within the same (i.e., *Guyparkeria*) genus in accordance with the definition (AAI of 65–95% and 16S rRNA of 95–98.6%) given in Konstantinidis et al. (2017) [22]. It is worthy to note that *Halothiobacillus* sp. S21.Bin061 shows the lowest ANI and AAI values among the 10 *Guyparkeria* genomes (Figure 2 and Figure 3). While clearly being a member of the *Guyparkeria* genes, it still seems to be evolutionarily more distant than the other genomes in this analysis. This fact, together with the lower completeness of this genome and the resulting lower CDS number (Table 1), leads to a bipartite pattern in the core genome development plot (see Section 3.3 for discussion). Given the described slightly higher deviation of strain S21.Bon061 from the other *Guyparkeria* genomes, it was used to root the phylogenetic tree. Due to the low completeness of *Halothiobacillaceae* bacterium SpSt-1134 and its relatively low AAI values (~68%) compared to other *Guyparkeria* genomes, it was omitted from the core genome and phylogenetic analysis.

It is noted that the genomes of *Halothiobacillus* sp. XI15 and *Halothiobacillus* sp. WRN-7 are of identical length (Table 1) and ANI values (Figure 3), although they are claimed to originate from different habitats and geographic locations (XI15 from Kebrit deep brine–seawater interface, Red Sea, Saudi Arabia and WRN-7 from saline-alkaline soil, Tianjin, China). As of the time of writing this manuscript, clarification has not been received from the author who submitted both of these genomes; as a result, no attempt was made to associate the observed phylogeny and genes with their origins.

The genomes of the type strain *G. hydrothermalis* R3, *Halothiobacillus* sp. SB14A and *G. halophila* sp2 share >97% ANI values (Figure 3), which satisfies the recommended criterion recommended for organisms of the same species (i.e., >95–96% ANI; [22,23]). Nonetheless, it warrants a careful look at their morphological and physiological differences before proposing their detailed taxonomic classification.

### 3.2. Phylogenetic Relatedness of Guyparkeria 16S rRNA Genes, Genomes and soxB Genes

The analysis of 16S rRNA genes, the gold standard for understanding phylogeny of life, indicated that the coastal strain *G. hydrothermalis* B1-1 is closely related to *Ha**lothiobacillus* sp. WRN-7 and *Halothiobacillus* sp. XI15, whereas the deep-sea vent chimney strain *G. hydrothermalis* R3 is closely related to marine sediment strain *Halothiobacillus* sp. SB14A and *G. halophila* sp2 from cold seep sediment. Congruent with the 16S rRNA gene tree, the two distinct clades are also formed with robust bootstrap support in the phylogenetic tree (Figure 4), which is in agreement with the observed ANI or AAI similarities. Similar clustering is also observed in the analysis of *sox*B genes (Figure 5), the marker gene for thiosulfate oxidation.

### 3.3. Pangenome of Guyparkeria

The analysis of the *Guyparkeria* genomes presented a pan genome size of 3204, with 46% (1458 CDS), 28% (909 CDS) and 26% (837 CDS) being in the core and accessory genomes, and as singleton genes, respectively (Appendix A). The core genome therefore accounted for ~63–77% of CDS in the studied genomes. This amount of shared genes in the eight studied *Guyparkeria* strains is quite high, when compared to that reported from clinical-relevant bacterial species from several phyla (~20–64% determined from 50 genomes per species; [24]) and the hydrothermal vent sulfur-oxidizing epsilonproteobacterial genera *Sulfurovum* (~5% from 20 genomes; [25]) and *Sulfurimonas* (~33–50% from 11 genomes; [26]).

From the *Guyparkeria* genomes, the Heaps’ Law function for the pangenome was determined as 2092*(N^0.185) (Figure 6A) and the exponential regression decay function for the core genome was 670.278*e^(−N/4.764)^ + 1384.351 (Figure 6B). The predicted alpha value of Heaps’ Law is 0.815, where a value of less than 1 indicates that the pan genome is open. It is understandable that the proportion of shared genes will decrease with the addition of more *Guyparkeria* genomes.

The core genome plot shows a bipartite pattern of values. This is due to the lower similarity of *Halothiobacillus* sp. S21.Bin061, as indicated by the lowest ANI and AAI values in the pairwise comparisons among the *Guyparkeria* genomes (Figure 2 and Figure 3). Another factor to explain this pattern is the smaller genome size of S21.Bin061 (92% completeness, Table 1), which has 140–380 CDS less than other *Guyparkeria* genomes (Table 1). The core development plot (Figure 6B) uses random sampling of sets of *n* genomes, with *n* being the range 1–10 in this case. Whenever the *Halothiobacillus* sp. S21.Bin061 genome is part of the random sample, the core genome size is ~250 genes smaller than when this genome is not sampled. If this genome is omitted from the analysis, the core development plot shows a narrower distribution of values (see Appendix A).

### 3.4. Genetic Basis of Key Metabolic Functions of Guyparkeria

The core genome provides the genetic basis of key metabolic features that have been reported for the four described *Guyparkeria* isolates, namely the type strain of *G. halophila* DSM 6132 (also being the type species of *Guyparkeria*; [4]), the type strain of *G. hydrothermalis* R3 [2], and strains SCN-R1 [7] and B1-1 [9].

Carbon assimilation by autotrophy. CO_2_ is fixed using the Calvin–Benson–Bassham (CBB) cycle (Figure 7 and Appendix A). To facilitate CO_2_ fixation, *Guyparkeria* employs protein-coated carboxysomes (encoded by *cso* genes) that house a carbonic anhydrase (CA), converting bicarbonate ions to CO_2_, and form I ribulose-bisphosphate carboxylase (RuBisCO), converting the 5-carbon compound ribulose-1,5-bisphosphate (RuBP) to form two molecules of 3-carbon compound 3-phosphoglycerate (3-PGA) [27]. It is believed that energy-dependent bicarbonate transporters, such as SulP, play a role in concentrating the intracellular bicarbonate level, similar to other autotrophs such as marine cyanobacteria [28] and sulfur-oxidizer *Thiomicrospira crunogena* [29]. Carbon-concentrating mechanism enables *Guyparkeria* to cope with times when the dissolved inorganic carbon availability is low or scarce.

The 3-PGA enters the CBB cycle that generates glyceraldehyde 3-phosphate (G-3-P), and regenerates RuBP to complete the CBB cycle. G-3-P is an important metabolic intermediate that generates precursors pyruvate and acetyl-CoA for biomass synthesis (e.g., lipids, nucleotides, proteins, etc.). A partial tricarboxylic acid (TCA) cycle is observed, with the genes encoding the enzymes of the oxidation arm being present in the *Guyparkeria* genomes but not that of the reductive arm (Figure 7). The incomplete TCA cycle explains why *G. hydrothermalis* R3 cannot assimilate acetate for heterotrophic growth [2]. G-3-P may also be directed to synthesize starch (Figure 7).

The described genetic composition of central carbon metabolism in *Guyparkeria* resembles very much that of sulfur-oxidizing bacteria *Thiomicrospira crunogena* XCL-2 [30]. Scott and coworkers [30] hypothesized that when there is ample supply of reduced sulfur compounds (electron donors) and oxygen (terminal electron acceptor), the CBB cycle is employed for growth, with some carbon being used to produce starch; when reduced sulfur compounds become limiting, starch is used as the main source for energy and carbon intermediates. This hypothesis of adaptive carbon use in response to changes in energy substrate levels may also apply to *Guyparkeria*. Scott and coworkers [30] also hypothesized that hydrogenase may play a role in maintaining a membrane proton potential in *Thiomicrospira crunogena* XCL-2, which is not likely to be applicable to *Guyparkeria*, because genes encoding for hydrogenase are not apparent in any of the *Guyparkeria* genomes (not among singleton genes (Appendix A) and no valid hit was found by hmmsearch). The absence of hydrogenase-coding genes explains the observation that *G. hydrothermalis* R3 does not grow under CO_2_ and H_2_/O_2_ [2].

Sulfur oxidation. *Guyparkeria* genomes encode a complete SOX pathway, as existing isolates showed complete oxidation of thiosulfate to sulfate (Figure 7 and Appendix A). Interestingly, the *sox* genes are not organized in a single operon, suggesting that in the absence of an alternative pathway, they are always expressed and do not require complex regulation [30]. The *sox* genes are split into three clusters: *sox*BAX, *sox*CD and *sox*ZY in all *Guyparkeria* genomes except that of SCN-R1 and S21.Bin061, in which the *sox*B gene is separated from the *sox*AX gene cluster. The production of sulfate results in lower pH, as has been reported for *Guyparkeria* isolates (an end point of pH 4.8–6.0; [1]). *Guyparkeria* cells grown under thiosulfate are coated with elemental sulfur. The accumulation of elemental sulfur outside of the cells could be explained by incomplete oxidation of thiosulfate at lower pH that inhibits the activity of subunit *sox*CD [31].

Multiple copies of genes related to sulfide dehydrogenases were identified. The presence of *fcc* and *sqr* genes encoding for flavocytochrome *c* (FCC) and sulfide:quinone oxidoreductase (SQR), respectively, suggested that they are responsible for the oxidation of sulfide to elemental sulfur with the involvement of electron transfer to cytochromes [32].

Thiocyanate oxidation. Of the four *Guyparkeria* isolates, it was reported that *G. halophila* DSM 6132 and *G. hydrothermalis* R3 do not use thiocyanate as an energy source, but only strain SCN-R1 [7]. Tsallagov et al. [6] revealed that SCN-R1 genome possesses the gene encoding for thiocyanate dehydrogenase as well as cyanate hydratase or cyanase (CYN). Interestingly, the *cyn* gene is present in all *Guyparkeria* genomes, and in its proximity is another copy of CA. Further investigation is needed to find out whether thiocyanate oxidation would be a general characteristics for *Guyparkeria*, and under what conditions.

Salt tolerance. A variety of compounds are known to be employed by diverse halotolerant or halophilic microorganisms to cope with osmolytic stress, such as ectoine in Proteobacteria and glycine betaine in halophiles [33]. As suggested for strain SCN-R1 [6], ectoine is considered to be the primary osmolyte for osmoregulation synthesized by halophilic *Guyparkeria* using *ect* genes (Appendix A). Removal of ectoine by *Guyparkeria* can be achieved by ectoine-degrading enzymes (encoded by *doe* genes). Secondary osmolytes, specifically glycine betaine, may be imported through membrane transporters in the ABC and BCCT families, including *Opu*D (Appendix A).

Other metabolisms. Ammonium is used as an N source [1], and is imported by ammonium transporter AMT. The addition of ammonium (0.8 g/L) was reported to shorten the lag phase of B1-1 culture when compared to adding nitrate and peptone of the same concentration [34]. Ubiquinone 8 (Q-8) is the dominant respiratory quinone used by *Guyparkeria* [1]. The *ubi* genes for ubiquinone 8 synthesis are present in all genomes. ATP is synthesized via oxidative phosphorylation, with genes encoding for complex I NADH:quinone oxidoreductase (NDH-1), complex III Cytochrome *bc*1 complex respiratory unit, complex IV Cytochrome c oxidase (including *cbb*3 type of high O_2_ affinity), complex V F-type ATPase being present in the *Guyparkeria* genomes (Appendix A). Multi-subunit cation transporters, such as that encoded by *Mnh* genes, are probably involved in Na+/H+ and pH homeostasis. *Guyparkeria* are capable of transporting essential inorganic nutrients and metal cofactors such as phosphate, molybdate/molydenate, zinc, iron, nitrate, cobalt, sulfate, manganese and copper (*cus* and *cop* genes) (Appendix A). Lipopolysaccharide production (*lpx* and *lip* genes) and exportation (*lpt* genes), as well as biofilm formation signaling genes, are present in most, if not all, *Guyparkeria* genomes, suggesting that *Guyparkeria* are capable of forming biofilm for protection against adverse conditions (e.g., toxins, heavy metals, fluid flow, etc.) [35,36]. In addition, *Guyparkeria* genomes possess genes encoding for the formation of type IV pili (T4aP, *pil* genes) (Figure 7). Type IV pili exhibit dynamic and diverse functions such as twitching motility, surface sensing, DNA uptake and even virulent infection to other cells [37], thus enabling *Guyparkeria* to respond to environmental changes.

### 3.5. Highlights on Some Accessory Genes in B1-1 and R3 Genomes

One of the physiological characteristics distinguishing *G. halophila* and *G. hydrothermalis* from *Halothiobacillus* is that tetrathionate was not a detectable intermediate when thiosulfate was provided as the sole electron donor for *Guyparkeria* cells, although they are able to oxidize tetrathionate [1]. It was reported that strain SB14A expressed *sox*B and *sox*C genes when growing on tetrathionate. Thiosulfate-oxidizing gammaproteobacteria that form tetrathionate either use *dox*DA-encoded thiosulfate:quinone oxidoreductase or *tsd*A-encoded thiosulfate dehydrogenase [38,39], yet these genes are not present in *Guyparkeria* genomes, except for a *tsd*A gene homolog that was identified in strain B1-1 and SCN-R1 (by automated genome annotation (Appendix A) and hmmsearch). Without empirical evidence, it is too early to comment on the actual function of the *tsd*A gene homologs encoded in the genomes of strains B1-1 and SCN-R1. It would be interesting to perform more targeted experiments to test whether any of the *Guyparkeria* isolates exhibits the physiological characteristic of converting thiosulfate to tetrathionate.

Interestingly, genes encoding for clustered regularly interspaced short palindromic repeats (CRISPR) are present in the genomes of strains R3, sp2 and SB14A, which form a cluster in phylogenetic trees constructed for 16S rRNA genes, and *sox*B genes and genome-wide shared genes. CRISPR genes are known as a bacterial defense mechanism against virus/phage infection. This can partly be explained by the considerable amount of viral load in marine environments (e.g., vent chimney, cold seep sediment and deep-sea sediment) from which these strains were isolated [40,41,42].

## Figures and Tables

**Figure 1 microorganisms-10-00724-f001:**
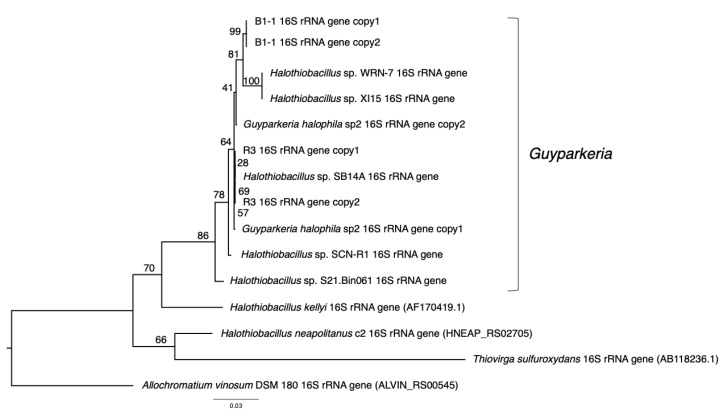
Phylogeny of *Guyparkeria* 16S ribosomal RNA genes. Tree topologies are supported by bootstrap values. Scale bar represents 0.03 changes in amino acid residues per position.

**Figure 2 microorganisms-10-00724-f002:**
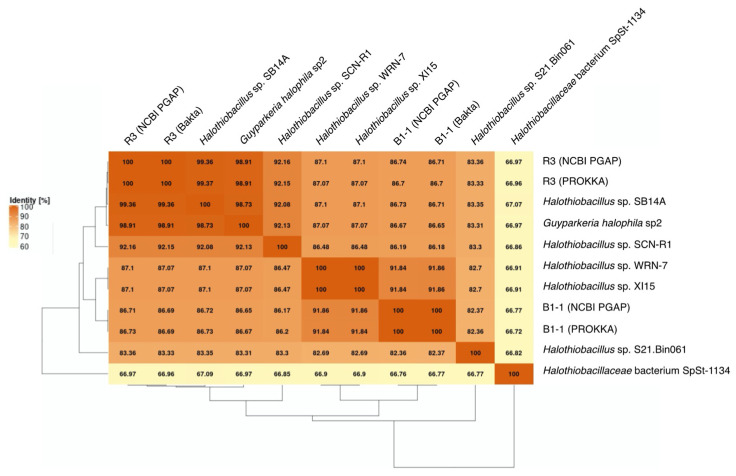
Average Amino Acid Identity (AAI) matrix. The heatmap with clustering is based on pairwise calculations. Percentage of identity is color coded for the range between 60% (cream) to 100% (orange).

**Figure 3 microorganisms-10-00724-f003:**
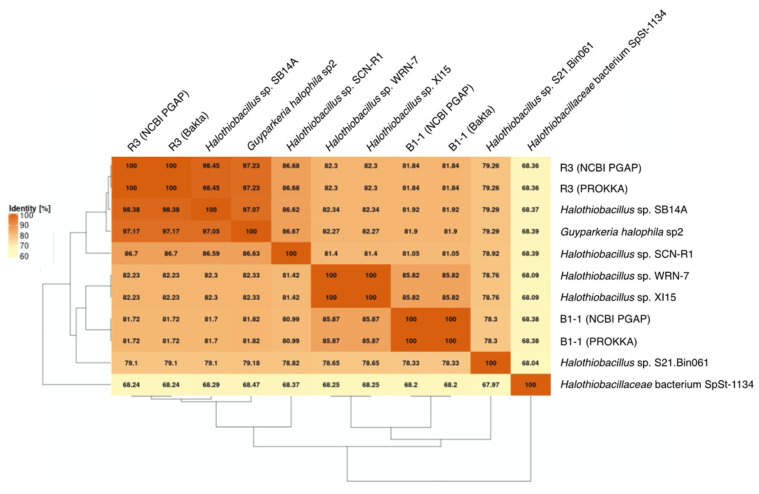
Average Nucleotide Identity (ANI) matrix. The heatmap with clustering is based on pairwise calculations. Percentage of identity is color coded for the range between 60% (cream) to 100% (orange).

**Figure 4 microorganisms-10-00724-f004:**
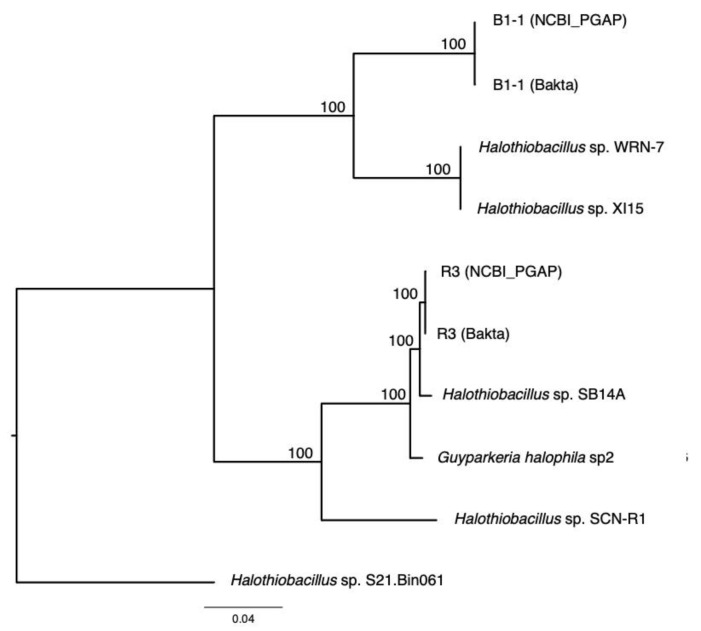
Phylogeny of *Guyparkeria* orthologous core genes. Tree topologies are supported by bootstrap values. Scale bar represents 0.04 changes in amino acid residues per position.

**Figure 5 microorganisms-10-00724-f005:**
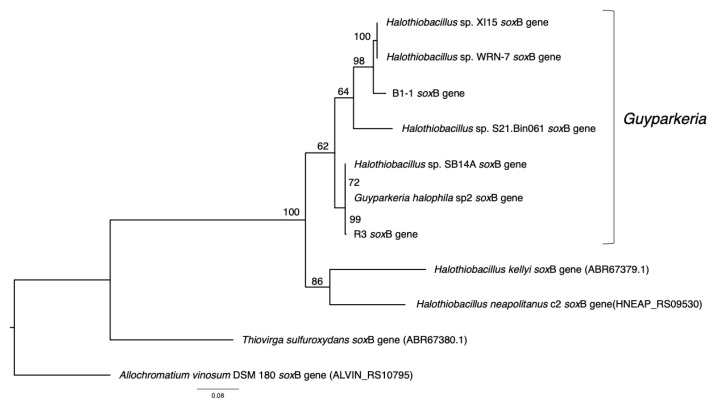
Phylogeny of *Guyparkeria sox*B genes. Tree topologies are supported by bootstrap values. Scale bar represents 0.08 changes in amino acid residues per position.

**Figure 6 microorganisms-10-00724-f006:**
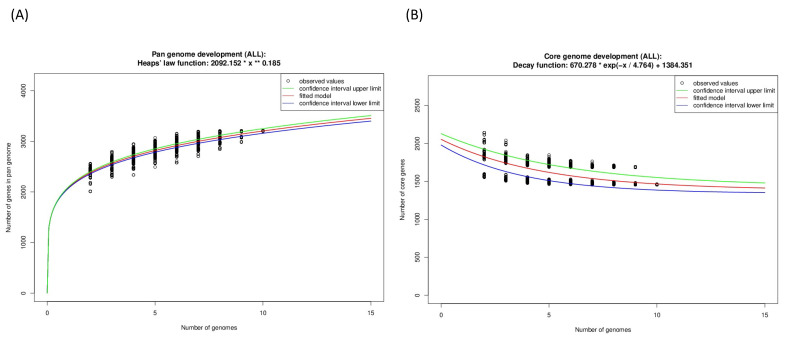
Development plots for *Guyparkeria* (**A**) pangenome and (**B**) core genome.

**Figure 7 microorganisms-10-00724-f007:**
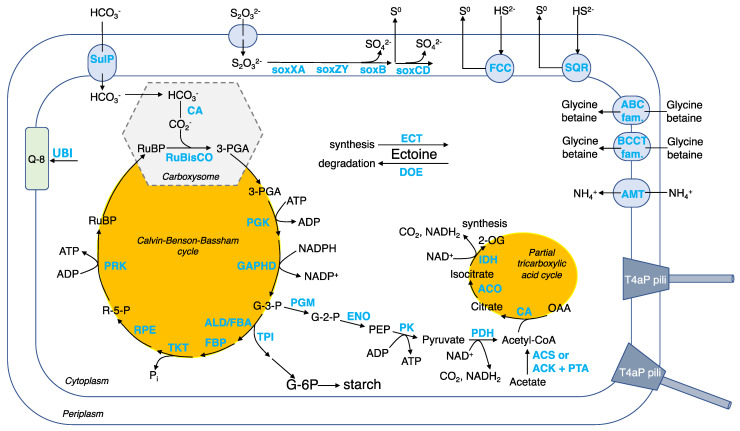
Schematic cell model for the genus *Guyparkeria*, with an emphasis on the carbon metabolism, sulfur oxidation, and salt tolerance. Abbreviation of metabolites (black font): RuBP, ribulose-1,5-bisphosphate; 3-PGA, 3-phosphoglycerate; G-3-P, glyceraldehyde 3-phosphate (also known as 3-phosphoglyceraldehyde); R-5-P, ribulose-5-phosphate; PEP, phosphoenolpyruvate; OAA, oxaloacetate; 2-OG, oxoglutarate; G-6-P, glucose-6-phosphate; Q-8, ubiquinone 8. Abbreviation of genes/enzymes (blue font): CA, carbonic anhydrase; RuBisCO, ribulose-bisphosphate carboxylase; PGK, phosphoglycerate kinase; GAPHD, glyceraldehyde-3-phosphate dehydrogenase; TPI, triose-phosphate isomerase; ALD/FBA, fructose-bisphosphate aldolase; FBP, fructose-bisphosphatase; TKT, transketolase; RPE, ribulose-phosphate 3-epimerase; PRK, phosphoribulokinase; PGM, 2,3-bisphosphoglycerate-independent phosphoglycerate mutase; ENO, enolase; PK, pyruvate kinase; PDH, pyruvate dehydrogenase; CS, citrate synthase; ACO, aconitate hyratase; IDH, isocitrate dehydrogenase; ACS, acetyl-CoA synthase; ACK, acetate kinase; PTA, phosphate acetyltransferase; SOX, SOX pathway for thiosulfate oxidation; FCC, flavocytochrome c-type sulfide dehydrogenase; SQR, sulfide:quinone oxidoreductase; ECT, ectoine synthesis enzymes; DOE, ectoine degradation enzymes; ABC fam., ABC transporter; BCCT fam., BCCT transporter; UBI, ubiquinone 8 synthesis enzymes.

**Table 1 microorganisms-10-00724-t001:** Brief summary of the nine genomes of the genus *Guyparkeria* used in this study.

NCBI Organism Name	*Guyparkeria hydrothermalis*	*Guyparkeria hydrothermalis*	*Guyparkeria halophila* *	*Halothiobacillaceae* bacterium	*Halothiobacillus* sp.^	*Halothiobacillus* sp. XI15 ^	*Halothiobacillus* sp. WRN-7	*Guyparkeria* sp. SCN-R1 ^	*Halothiobacillus* sp. SB14A ^
Strain	B1-1	R3	sp2	SpSt-1134	S21.Bin061	XI15	WRN-7	SCN-R1	SB14A
Habitat	Sediment in marine cage culture area, Fujian, China	Hydrothermal vent chimney, North Fiji Basin, Pacific Ocean	Sediment in cold seep	Hot spring sediment, California, USA	Lacustrine sediment, Tibet, China	Kebrit deep brine-seawater interface, Red Sea, Saudi Arabia	Saline-alkaline soil, Tian Jin City, China	Thiocyanate-degrading bioreactor, Eerbeek, Netherlands	Ocean sediment, Arabian Sea Oxygen Minimum Zone, India
Isolate?	Yes	Yes	Yes	No	No	No	No	Yes	Yes
NCBI BioSample ID	SAMN23673796	SAMN23673797	SAMN13381662	SAMN09639045	SAMN13520459	SAMN04318430	SAMN04419354	SAMN10095268	SAMN11475377
NCBI Assembly ID			GCF_009734265.1	GCA_011380105.1	GCA_011389965.1	GCF_001469965.1	GCF_001641825.1	GCF_003932495.1	GCF_005096345.1
GTDB species representative	-	-	yes	yes	yes	-	yes	yes	-
Number of scaffolds	1	9	1 (complete genome)	235	130	14	14	42	62
Number of bases (nt)	2392942	2433989	2594469	1188407	2016313	2291306	2291306	2406866	2428842
Completeness (%)	100	100	100	68.91	92.53	100	100	100	99.43
Contamination (%)	0	0	0	1.15	0	0	0	0	0
GC content (%)	65.2	66.5	66.2	52.4	63.9	66.2	66.2	64.7	66.5
Number of rRNAs (5S, 16S, 23S)	2, 2, 2	2, 2, 2	2, 2, 2	0	1, 1, 1	1, 1, 1	1, 1, 1	1, 1, 1	2, 1, 1
Number of tRNAs	47	46	50	14	40	45	45	49	44
Number of coding sequences CDS (predicted by CheckM)	2172	2142	2325	1222	1943	2081	2081	2180	2207
Number of coding sequences CDS (predicted by PROKKA)	2168	2141	2320	1033	1875	2067	2067	2170	2172

Note: * *Guyparkeria halophila* is the type species of *Guyparkeria* according to Boden (2017) [1]. ^ Scaffolds < 1500 nucleotides (nt) were removed in this study.

## Data Availability

BioProject PRJNA786470 (BioSample ID SAMN23673796 for B1-1 and SAMN23673797 for R3).

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
