# Peer review of "Metabolism of the Genus *Guyparkeria* Revealed by Pangenome Analysis"

_microorganisms, 2022, doi:10.3390/microorganisms10040724_

Round 1

Reviewer 1 Report

The work presented by the authors is of particular interest for a detailed knowledge of the genus Guyparkeria. The genomic approach provides substantial support to experimental evidence. The study carried out seems well organized and therefore suitable for publication in this journal.

Reviewer 2 Report

The authors provide a whole genome-based comparison of nine (or actually eight) genomes of Guyparkeria. The authors sequenced two of the genomes and made a comparison to the other seven genomes already available in the database in an attempt to compare the metabolic capabilities and adaptations of these species. However, in general the study would benefit from a more in-depth comparison and description of unique features of this genus. The provided metabolic analysis appears to be a general overview of these (already known) pathways without substantial information on the differences in the genus from other genera. The study would benefit from a deeper analysis and less speculation before it is acceptable for publication.

Also, a general check of the English spelling and writing would be helpful.

Several points need to be addressed before this is suitable for publication:

  • Lines 116-118: are these abbreviations for the 40 selected genomes? If so,they should use accession numbers (and species names), or it should be explained what these are.
  • Table 1 is very hard to read in its current form. This should be reformatted (shortened maybe?) to make it legible in publication).
  • Lines 191-192: how many differences were there in the 16SrRNA fragments (out of how many nt compared)?
  • Figure 1: not all the branches appear to have bootstrap values. Also, it might be better to include the actual bootstrap value numbers instead of the half and full circles. The 75% and 95% are arbitrary and it should be explained why these were chosen as cut off values.
  • Figures 4 and 5. Same comments as for Figure 1. In addition,the black background makes it nearly impossible to see the bootstrap circles. Please reconfigure with a white or clear background and all bootstrap values as text.
  • Line 248: ‘say 15 in total…’ How is this estimated? This seems a bit of a random guess? Please explain why 15 was stated here.
  • Figure 6B: These fits do not appear to be the best fits for these data sets. Is there any statistical significance (at least Rsquare values or so) that can be added to validate these fitted curves. And justify why this fit was chosen for these datapoints.
  • Lines 301-303: how was the lack of hydrogenase genes confirmed? Was any additional BLAST or PCR performed to confirm the automated annotation?
  • Line 308-309: How can it be concluded from genomic data alone that these genes are always expressed? At the very least some transcriptomics data would be needed to confirm this. This conclusion is too speculative and should be removed.
  • Lines 316-318: The presence of these genes does not suggest that they are responsible for the oxidation of sulfide. The assumed function of these genes is based on previous studies of these genes and their protein products and references for this should be cited here.
  • Lines 329-331: these statements need references. It is also not clear how these relate to the Guyparkeria genomes. Is it suggested that the species all use both ectoine and glycine betaine as osmolytes? If so, what evidence is there (possibly from other genera...)
  • Lines 361-365: how was the lack of these genes confirmed? Was any additional BLAST or PCR performed to confirm the automated annotation? Perhaps these were not annotated correctly, or hypothetical proteins in the automated annotation.
  • Line 364-365: that line is too speculative and should be removed as there currently is not evidence for this provided.

Minor points that should be addressed:

- line 20: ‘specie’ should be ‘species’ (the word specie is not just the singular form, but as a different meaning)

- line 25: delete ‘are’

-line 31: should be ‘centered around’

-line 57: better to say ‘…was filtered twice to sterilize the solution…’

-line 64: ‘autoclave’ should be ‘autoclaving’

-line 65: ‘multiple transfers’ (plural)

-line 215: ‘it warrants’ (space)

-line 250 and 251: Guyparkeria should be in italics

-line 302: change to ‘apparent’

-line 324: ‘the cyn gene’ add ‘the’

-line 330: italicize ‘doe’

-line 333: the subheader ‘Others’ seems somewhat weird here. Maybe replace it with ‘other metabolism’ or similar.

-line 354: ‘genomes’

- line 368: ‘trees’

- line 369: ‘CRISPR’, not ‘CRSPR’

- Species names throughout the references should be in italics.

Round 2

Reviewer 2 Report

The authors did a good job of revising the manuscript and addressing the previous concerns. However there are still some minor concerns that should be addressed:

  • Line 20: This sentence should either be '...Guyparkeria species, previously were classified' (not 'was'), or be rephrased as, 'the species Guyparkeria was previously classified'.
  • Line 32 needs a reference
  • Line 52: did the media really have 0.23 g CaCl2 (and additional CaCl2 in the trace elements)? That is an unusual high amount of Ca, but maybe this is just a typo. 
  • Line 79: Sanger should be capitalized
  • Line 204-205: the authors never explain why there are two different copies or 16SrRNA in the sp2 genome. Is the genome a mixture of two populations, or has this been observed in other strains as well? 
  • Line 223: the half-filled and filled circles reference should be removed as they are no longer in the figure. Same with lines 254 and 257.
  • Figures 2 and 3: the color legend on the side is too small. Please increase or explain the colors in the figure legend. 
  • Line 283: 'genome should not be in italics
  • Lines 295 and 305: it is the CBB cycle, not CCB cycle

Figure 5 still shows up with a black background in my PDF version, but hopefully that will be fixed in the final editing. 
